# The Efficacy of Cannabis on Multiple Sclerosis-Related Symptoms

**DOI:** 10.3390/life12050682

**Published:** 2022-05-05

**Authors:** Fatma Haddad, Ghadeer Dokmak, Rafik Karaman

**Affiliations:** 1Pharmaceutical Sciences Department, Faculty of Pharmacy, Al-Quds University, Abu Dis, Jerusalem P144, Palestine; iamfromhebron@hotmail.com (F.H.); ghadeer_88@live.com (G.D.); 2Faculty of Life Sciences, University of Bradford, Bradford BD7 1DP, UK; 3Department of Sciences, University of Basilicata, 85100 Potenza, Italy

**Keywords:** *Cannabis sativa*, marijuana, cannabinoids, 9-tetrahydrocannabinol (THC), multiple sclerosis (MS)

## Abstract

Multiple sclerosis (MS) is known as an autoimmune disease that damages the neurons in the central nervous system. MS is characterized by its most common symptoms of spasticity, muscle spasms, neuropathic pain, tremors, bladder dysfunction, dysarthria, and some intellectual problems, including memory disturbances. Several clinical studies have been conducted to investigate the effects of cannabis on the relief of these symptoms in MS patients. The efficacy of *Cannabis sativa* (*C. Sativa*) in the management of MS outcomes such as spasticity, pain, tremors, ataxia, bladder functions, sleep, quality of life, and adverse effects were assessed in this review. Most clinical studies showed the positive effects of cannabinoids with their different routes of administration, such as oromucosal spray and oral form, in reducing most MS symptoms. The oromucosal spray Nabiximols demonstrated an improvement in reducing MS spasticity, pain, and quality of life with a tolerated adverse effect. Oral cannabinoids are significantly effective for treating MS pain and spasticity, while the other symptoms indicate slight improvement and the evidence is quite inconsistent. Oromucosal spray and oral cannabis are mainly used for treating patients with MS and have positive effects on treating the most common symptoms of MS, such as pain and spasticity, whereas the other MS symptoms indicated slight improvement, for which further studies are needed.

## 1. Introduction

MS is a neurological disease with an autoimmune origin that affects and damages the central nervous system and affects 2.3 million people worldwide [1,2]. This demyelinated disease leads to severe impairment of nerve signal transmission between the brain and spinal cord that causes a loss of myelin sheath [1,2].MS has been characterized by symptoms of spasticity, muscle spasms, tremors, bladder dysfunction, neuropathic pain, dysarthria, and some intellectual problems, including memory disturbances [3,4]. Some drugs have been licensed to slow down disease progression and reduce relapse frequency [3]. However, more studies are needed to further alleviate the disabling symptoms of MS patients.

In the 21st century, the Sativa plant has become the most widely used illicit drug [5]. It isalso known as hemp, cannabis, or marijuana, and comes in a variety of forms, including cigarettes or hash pipes and sweets or brownies [5,6]. The cannabis plant, which includes over 560 identified components, is primarily composed of phytochemicals [5]. The two prominent species of Cannabis plant, which are mostly used for recreational and medical, are *Cannabis indica* and *Cannabis sativa* (*C. Sativa*) [7,8]. Both plants are composed of different cannabinoid compositions of THC and CBD [9,10]. Previous research hasshown the different effects of each species due to the different concentrations of the two main components [11]. *C. Sativa* has a higher ratio of CBD to THC and the reverse is seen for *C.indica* [10]. There are about 100 cannabinoids in *Cannabis sativa* [12], the most well-known of which are 9-tetrahydrocannabinol (THC) and cannabidiol (CBD) (Figure 1a,b). The endogenous cannabinoid system, which includes CB1 and CB2 receptors, is where cannabinoids get their effects. The psychotropic effects of THC are primarily mediated by a CB1 receptor agonist actions. CBD, on the other hand, is hypothesized to bind to CB1 and CB2 receptors and act as an antagonist [5,13].

Cannabis has been used for over 5000 years, with the discovery of the endogenous cannabinoid system occurring more than a decade ago. CB1 and CB2 are cannabinoid receptors that are linked to adenylyl cyclase negatively and mitogen-activated protein kinase positively through the Gi/o protein [14,15]. CB1 and CB2 are distributed in the central and peripheral nervous systems and immune systems [16]. CB1 receptors are found mostly in nerve terminals in the central nervous system and some peripheral tissue and are linked to a specific type of calcium and potassium channel via a G protein. Because it blocks pain pathways in the brain and spinal cord, the CB1 receptor’s main function is to suppress neurotransmitter release, and it plays a crucial role in mediating the pain-relieving effect of cannabis [17].

Phytocannabinoids, endocannabinoids, and synthetic cannabinoids are the three types of cannabinoids. The major chemical components of cannabis are phytocannabinoids, which include a variety of non-cannabinoid C21 terpenes phenolic compounds, or C22 for the carboxylate group, which is largely synthesized in cannabis [18,19]. Heat can decarboxylate phytocannabinoids, which are biosynthesized in carboxylated form [20].

Cannabis isconsidered a promising anti-inflammatory and immunosuppressive agent due to its central and peripheral actions on CB1 and CB2 receptors that mediate different intracellular pathways when activated [21]. In addition to the effects of the cannabinoids on CB1 and CB2 receptors, cannabinoids have effects on nuclear receptors and ion channels by their activity on other transmembrane G protein-coupled receptors (GPCRs) and have a modulating activity on opioid and serotonin receptors [16]. THC is a psychoactive component of cannabis with the highest potency. THC can be utilized to treat neuropathic and chronic pain because of its intoxicating, anti-emetic, and anti-inflammatory properties [22]. CBD does not cause psychosis, although it does have pharmacological effects on pain and spasticity [23].

The main psychoactive constituent of the *C. Sativa* plant, 9-THC, was discovered in the late 1980s and was shown to have activity on a specific cannabinoid receptor in the brain (the cannabinoid CB1 receptor), which had a huge impact on the development of cannabinoid therapeutic drugs and their potential to relieve MS spasticity symptoms [24]. Preclinical animal studies of MS suggested that cannabinoids have anti-septic effects due to the activation of CB1 receptors, which inhibit the release of classical neurotransmitters, such as glutamine, while also decreasing neuronal excitability by activating somatic and dendrite potassium channels [25,26].

CBD is a key non-psychotropic cannabinoid present in *C. Sativa* that accounts for up to 40% of the cannabis plant’s extract and binds to a wide range of physiological targets of the endocannabinoid system in the human body. CBD has shown potential in the treatment of MS symptoms. CBD, in particular, has been shown in numerous trials to reduce stiffness, discomfort, inflammation, exhaustion, and depression in MS patients, resulting in increased mobility [27,28]. CBD also affects the non-cannabinoids receptors GPCRs and ion channels that will lead to pain regulation and anti-inflammatory effects through receptor modulation [16].

Several attempts have been made to determine the primary genetic factor contributing to MS progression and severity [29,30,31]. Although it was demonstrated that genetic effects on MS disease severity and susceptibility are polygenic with modest influence [30,31], more studies are needed to further understand the pharmacogenetics of MS disease and to define the molecular target of CBD in MS patients by performing the ex vivo/in vitro research in human immune cells as reported by Furgiuele et al. [31].

The efficacy of oromucosal spray nabixiomol, oral dronabinol, and oral nabilone forms of cannabis on MS-related spasticity, pain, tremor, urine function, sleep disturbances, quality of life, disability, and disability progression is assessed in this review. The sections below go through the signs and symptoms of MS, as well as the treatment options.

### 1.1. MS-Related Symptoms

#### 1.1.1. Spasticity in MS

Spasticity is a common symptom of MS, affecting 60–84% of patients, particularly throughout the disease’s progression [32]. Spasticity is caused by injury of the higher corticosteroids motor neurons, as well as aberrant supraspinal driving of spinal reflexes [33], which can exacerbate other MS symptoms and harm the patient’s quality of life [34,35,36].

MS spasticity is a complicated condition characterized by the most prevalent symptom, muscle rigidity, which is induced by the stretch reflex’s hyperexcitability. Fatigue, discomfort, and bladder dysfunction are among the more prevalent and troubling symptoms [37]. Combining nonpharmacological and pharmacological therapies is the most commonly employed treatment [38]. Increasing severity of MS spasticity leads to decreased patient satisfaction, with limited and ineffective oral pharmacotherapy available to treat MS spasticity [39].

MS spasticity therapy is used to improve functional ability, assist rehabilitation, avoid contractures, and relieve discomfort in people with MS. The only use of cannabinoids in neurological disorders and the only complementary medicine intervention with high-level evidence for efficacy in MS is pharmaceutical cannabinoids for spasticity, according to most recent studies [40,41].

The most prevalent and available treatment for MS spasticity is multimodal [38], which combines non-pharmacological and pharmaceutical therapies. The majority of current therapy approaches result in patient dissatisfaction. A new cannabis-based pharmaceutical option with different administration methods has been created. Surprisingly, most studies revealed that topical cannabis was most commonly used for treating MS spasticity because of its pharmacokinetic qualities, although smoking or vaping was the most popular strategy in other studies [42,43]. THC and THC: CBD is the most commonly used cannabinoid to treat spasticity, according to reviews by Ben Amar, Koppel et al., Lakhan, Rowland, and Krast et al. [40,44,45,46].

#### 1.1.2. MS-Related Pain

MS is frequently linked to pain [47]. Pain has long been recognized as a significant element in MS patients’ overall health-related quality of life (HRQoL). Patients with MS who have pain have reported decreased HRQoL, as well as physical and emotional dysfunctions [48,49]. Neurological injury, as well as neurological dysfunction and incapacity, might be the source of severe discomfort [50]. Continuous central neuropathic pain, intermittent central neuropathic pain, musculoskeletal pain, and mixed neuropathic and non-neuropathic pain are all examples of pain associated with MS [49]. Despite significant advancements in providing novel therapeutic techniques for the symptomatic treatment of MS, such as pain, the present therapeutic modalities are insufficient to meet the needs of patients [50,51].

#### 1.1.3. MS-Related Tremor and Ataxia

MS affects the cerebellum and its efferent and afferent pathways, causing tremors, ataxia, and dysarthria in both acute and chronic symptoms [52]. More than 80% of MS patients have tremors or ataxia at some point during their illness [53]. Several pharmacotherapies for the management of tremors or ataxia associated with MS are available, but they are often ineffective [52,54]. Because of the absolute and comparative efficacy and acceptability of pharmaceutical treatments are inadequately recorded, a Cochrane review of six randomized controlled studies for the management of ataxia in MS was unable to provide any recommendations [55]. Furthermore, several randomized controlled studies have looked at the effect of cannabis extracts on these symptoms, but they have concluded that cannabinoids are useless in treating MS tremor and ataxia [40,55,56,57]. As a result, it is prudent to seek out novel treatments.

#### 1.1.4. Bladder Dysfunction in MS

In MS patients, lower urinary tract dysfunction is a prevalent symptom. It usually appears in the later stages of the disease. In total, 50–90% of MS patients develop this symptom after 6 years of disease [58,59,60], which is primarily attributable to a neurogenic overactive bladder [61]. Urinary urgency and urge incontinence can hurt a patient’s quality of life and cause immobility. Anti-cholinergic and intermittent self-catheterization may be beneficial in the early stages [62]. A total of four reviews looked at the effect of cannabis on MS bladder function symptoms [40,44,46,63].

#### 1.1.5. MS-Related Sleep Disorders

One of the most prevalent symptoms among MS patients is sleep disturbances [64]. The most common causes of MS sleep problems are multifunctional, and they are linked to immunotherapy and symptomatic therapies, as well as MS-related symptoms such as pain and exhaustion [65]. Patients with untreated sleep disturbances are at a higher risk of developing diseases, which can have a long-term impact on their health [66,67,68]. Three studies looked into the impact of cannabis on MS sleep problems. However, no conclusion was reached regarding the clinical effects of cannabis in MS patients with sleep difficulties [44,46,63].

#### 1.1.6. Health-Related Quality of Life in MS Patients

HRQoL (Health-related Quality of Life) is a multidimensional perception that encompasses physical, emotional, mental, and social functioning [69]. MS patients have been found to have a lower HRQoL than the general population. The detrimental influence of illness symptoms on daily living performance could explain the worse HRQoL associated with MS [70,71]. In MS patients, the influence of cannabis on HRQoL was investigated, with inconsistent results [46,54,72].

#### 1.1.7. MS-Related Disability and Disability Progression

Progression MS is a disability-causing disease that is characterized by relapsing-remitting neurological episodes [73]. Progression is linked to inadequate treatment and diagnosis, which results in demyelination, axonal and neuronal loss, and various cognitive, sensory, and motor problems [73,74]. It occurs as a result of central nervous system lesions and is driven by central inflammatory and other neurodegenerative effects that underpin irreversible disability. Many studies looked at the impact of cannabis on MS impairment and signs of disability progression [46,63,72,75]. The majority of these reviews did notfocus on MS disability, and it was treated as a secondary result with no conclusions about cannabinoids’ impact on it.

## 2. Cannabinoid Agents for the Treatment of MS

### 2.1. Nabiximols

Nabiximols is the generic name for Sativex^®^, an oromucosal spray containing a 1:1 molecular ratio of THC and CBD (Nabiximols (c), Figure 1) [76]. It has been licensed in several countries for the treatment of severe spasticity in MS patients [5]. The THC, CBD, and the small number ofother constituents of the plant extract, including other cannabinoids and terpenoids dissolved in ethanol, make up around 70% of the constituents in Nabiximols [76]. There are 2.7 mg, 2.5 mg, and 0.04 g of THC, CBD, and ethanol, respectively, in each dose of oromucosal spray. Its administration has a favorable pharmacokinetic profile, with fewer first-pass effects and a low plasma concentration, resulting in the avoidance of psychoactive effects that are caused by smoked cannabis [76,77,78].Furthermore, its maximum plasma concentration would rise gradually within 2–4 h after administration, while the quick onset effect will appear after 15–40 h, making treatment adjustments easier [77].The synergistic interaction is based on a low-dose combination of THC and CBD, which results in reduced euphoric effects and improved cannabinoid-mediated anti-spasticity therapeutic benefits [76]. According to much research, starting therapy with a 14-day dose titration period is recommended to reach up to 12 sprays as the highest dose per day [78]. Individual dose distribution across a day, as well as dose change over the treatment course, is possible depending on the intensity of the disease. The recommended maximum daily dose is 12 sprays with at least 15 min between each spray [77]. Nabiximols were tested in MS patients with a variety of symptoms, such as stiffness, pain, tremor, and bladder function, in several clinical investigations [56,79,80,81]. Nabiximols were found to have good effects on spasticity, pain, and quality of life in MS patients in the majority of trials (Table 1) [56,79,80,81].

#### 2.1.1. Effects of Nabiximols on MS-Related Spasticity

Wade and colleagues reported an open-label extension trial in 2006 that included 137 patients diagnosed with MS after 6-week placebo-controlled research, in which the efficacy and tolerability of Sativex for the management of spasticity and other symptoms were assessed for about 15 weeks [82]. The primary outcome was assessed using a visual analog scale, which revealed a consistent reduction in spasticity. Furthermore, it had unfavorable effects on MS pain, tremor, and bladder symptoms, and 58 patients dropped out due to ineffectiveness [82].

Collin et al. conducted a 6-week, double-blind, placebo-controlled research on 189 patients in 2007, comparing the effects of Sativex between the groups [83]. There was a significant difference in spasticity reduction between the active and placebo groups, with a decrease in the numerical rating scale (NRS) score of *p* = 0.048 [83]. As a secondary endpoint, however, there was no therapeutic impact among other Ashworth scores [83].

Novotna and colleagues conducted a large multicenter phase III trial in 2011 that included two phases of study design: phase A began with a four-week single-blind treatment phase to identify early responders to oromucosal spray Nabixomols, followed by phase B, which was a 12-week randomized, double-blind study phase [77]. The recommended highest daily dose of nabiximols was 12 sprays. In a study comparing oral mucosal spray to placebo, the oral mucosal spray was linked to a 51% reduction in MS spasticity. In addition, the NRS ratings decreased by 20% [77].

Based on a series of randomized controlled clinical trials vs. placebo, Nabiximols were given and licensed for their therapeutic efficacy and relief of spasticity-related symptoms [56,79,83]. The first line anti-spastic study was recently designed and implemented using an enriched–design methodology as proof of the German authority’s request that add-on Nabixomolsweremore effective than readjusting the anti-spasticity medication regimen alone in providing symptomatic relief of MS spasticity during a 4-week trial period compared to placebo [84]. SAVANT, a well-designed research of oromucosalNabixomol as an add-on therapy for the treatment of MS spasticity in MS patients, found that it had a promising effect in reducing moderate to severe spasticity symptoms [85].

Nabiximols are approved and advised as an add-on treatment for adult patients with moderate to severe resistant MS spasticity in several Western countries [86,87].

#### 2.1.2. Effect of Nabiximols on MS-Related Pain

Nabiximols (oromucosal spray): randomized, placebo-controlled research in which 66 patients were randomly assigned to Nabiximols or placebo found that Nabiximols was effective in alleviating both central pain associated with MS and pain-related sleep disruption [88]. In a different double-blind experiment conducted in 2013, 339 patients with central neuropathic pain associated with MS were randomized to 167 Nabiximols and 172 placeboes, with the Nabiximols showing a statistically significant response rate at week 10 compared to the placebo [80]. The authors explained the findings by stating that their patient population may represent a particularly treatment-resistant group because they had long-standing pain spanning more than 5 years on average, and they demonstrated that those with a less than 4-year history of neuropathic pain were more likely to be resistant to treatment [80]. Nine randomized controlled studies with 1289 individuals were included in a recent systematic meta-analysis of THC: CBD oromucosal spray and placebo for the treatment of chronic neuropathic pain. Nabiximols were found to relieve chronic neuropathic pain more effectively than placebo, with a small effect size [89]. This suggests that further research into the full potential of Nabiximols in these people could be very interesting.

#### 2.1.3. Effects of Nabiximols on MS-Related Tremor

Only 13 of the individuals in a Class I trial using the Nabiximols to see if cannabis extracts had an effect on symptoms in MS had a tremor as their predominant symptom. The Visual Analogue Scale did not change when (THC/CBD) oral spray was used at a dose of 2.5/120 daily in divided doses. Due to the small number of participants, no conclusion or difference was reached [56]. A more recent investigation used a 0–10 numeric rating scale (NRS) to assess the severity of MS symptoms in 337 patients treated with Nabiximols, with an undetermined number of patients rating tremor on the NRS, and found no difference between Nabiximols and placebo [79]. Nabiximols were mentioned in a systemic review published by Koppel et al. in 2014. For treating tremors associated with MS, Nabiximols were deemed “potentially ineffective” [40]. As a result, it appears that Nabiximols is unsuccessful in the treatment of MS-related tremors and ataxia.

#### 2.1.4. Effect of Nabiximols on MS-Related Bladder Function

THC: CBD oromucosal spray (Nabiximols) reduced urine frequency of nocturia and improved incontinence-related quality of life measures in Class 1 research [90]. Nabiximols were found to have a minor effect on the alleviation of bladder dysfunctions in MS patients [90].

The impact of a cannabis-based medicinal extract on MS symptoms was studied in 160 MS patients in a double-blind, randomized, placebo-controlled class 1 study. Sativex’s effect on the symptoms of MS bladder dysfunction was investigated [56]. These groups did not experience any improvement in bladder difficulties after using Sativex [56]. Another randomized controlled research [90] looked at the efficacy and safety of Sativex in treating MS bladder dysfunction symptoms in 135 MS patients. Sativex was utilized to treat detrusor overactivity MS symptoms as an add-on therapy [90]. The primary endpoint of this double-blind, randomized, placebo-controlled parallel trial after 10 weeks revealed no significant difference between Sativex and placebo in the mean daily bouts of incontinence (*p* = 0.056) [90]. Sativex was found to be successful in four secondary endpoints out of seven, with a significance of *p* = 0.00. Sativex appears to have some effect on improving bladder dysfunction in MS patients; however, the primary endpoint did not reach statistical significance [90]. Giorgia et al. conducted a pilot prospective trial in 2017 to examine the effect of THC/CBD oromucosal spray on bladder dysfunction symptoms in MS patients. A total of 21 MS patients were examined, with 15 of them receiving a detailed clinical evaluation (extensive urodynamic studies) before and after one month of oromucosal THC/CBD treatment [91]. Overactive symptoms were reported to be reduced (*p* = 0.001), and an oromucosal spray administration was found to be beneficial in treating overactive bladder symptoms. THC/CBD oromucosal spray can improve overactive bladder in people with MS-resistant symptoms, according to a study [91].

#### 2.1.5. Effects of Nabiximols on MS-Related Sleep Disturbances

Russo et al. conducted a review of utilizing Sativex to examine sleep disorders and found that there was a sleep improvement with no medication tolerance [92]. Wade et al. assessed the efficacy of Nabixomols on 160 participants in a double-blind, randomized, placebo-controlled, parallel-group research in 2004 [56]. The trial lasted 6 weeks, followed by a 4-week active extension period. The administration of Nabixomols resulted in a considerable improvement in sleep difficulties in this study [56]. In 2011, Novotna published the results of a pivotal phase three trial for Nabixomols, which revealed a significant improvement (*p*-value 0.0001) in MS sleep problems, which were considered a secondary endpoint in the study [77].

#### 2.1.6. Effects of Nabiximols on MS-Related HRQoL

The effects of Nabiximols on MS-specific HRQoL were studied for 3 to 4 months in the MOVE 2 research [93]. Although the MOVE 2 trial found a beneficial effect of Nabiximols on physical and mental HRQoL, researchers found no change in HRQoL in MS patients when they assessed them for a year using the same method as the MOVE 2 study [93,94]. In total, 337 MS patients were evaluated in a 15-week randomized controlled trial who reported some favorable but statistically insignificant effects in HRQoL with Nabiximols therapy [79]. Another 19-week trial that treated 241 MS patients with Nabiximols revealed similar negligible improvements in HRQoL [77].

#### 2.1.7. Effects of Nabiximols on MS-Related Disability and Disability Progression

In terms of disability and progression, evaluations found no significant changes in various indexes, such as the Barthel index of daily living activity and walking time of MS disability [56,63,95]. The CUPID experiment assessed the impact and safety of oral cannabis THC to delay progressive inflammatory brain illness in MS patients (CUPID) [96]. It was a randomized, double-blind, placebo-controlled, parallel-group, multicenter trial. A total of 498 participants were randomized (332 to active and 166 to placebo) and 493 (329 actives and 164 placebos) were examined in this study. Patients were given either (9)-THC or placebo in a 2:1 ratio. There was no significant treatment impact in either the primary or secondary outcomes. Furthermore, there were no substantial safety issues, but unfavorable side effects appeared to have an impact on compliance [96].

### 2.2. Dronabinol

The primary THC isomer present in the cannabis plant, (−)-trans-9-THC(Dronabinol(d), Figure 1), is known by the generic name Dronabinol. Dronabinol is available in three strength formulations: oral soft gelatin capsule, oral soft gelatin capsule, and oral soft gelatin capsule (2.5, 5, and 10 mg). It was first used to treat chemotherapy-induced vomiting and nausea, as well as anorexia and weight loss in AIDS patients, and it was tested in ten clinical trials for its efficacy and safety in treating MS symptoms including spasticity, pain, tremor, bladder function, sleep, quality of life, and side effects [57,97,98,99,100,101,102,103,104,105]. Dronabinol appears to be highly effective in the treatment of pain in MS patients, according to the bulk of clinical evidence, although other symptoms showed only little improvement (Table 1) [95].

#### 2.2.1. Effects of Dronabinol on MS-Related Spasticity

Zajick et al. conducted a multicenter, randomized, placebo-controlled research on 630 patients aged 18 to 64 in 2003 [57]. The effect of orally administered Marinol (Dronabinol) and Cannador on MS symptoms were studied for 15 weeks in this design study [57]. Weight-adjusted 0.25 mg/kg 9-THC (Dronabinol, supplied orally in 2.5 mg capsules), natural Cannabis oil (Cannador), or placebo were used to provide the dose [57]. The study assessed the effects of orally delivered cannabis extract (Cannador), Dronabinol, or placebo on spasticity following 28 days dose-titration phase using the Ashworth scale. Over a year, 80% of the subjects demonstrated a moderate reduction in spasticity [57]. There was no significant therapeutic impact between the active and placebo groups in the primary outcome. In comparison to the placebo group, the groups who took Dronabinol or Candor in spasticity demonstrated better improvement in the secondary result. They discovered that subjects who took Dronabinol exhibited a slight improvement on the Ashworth scale (*p* = 0.003) after 12 months of follow-up, which was not detected in the canned or placebo groups [57].

#### 2.2.2. Effects of Dronabinol on MS-Related Pain

Schimrigk et al. published a study that found Dronabinol to be superior to placebo in the treatment of neuropathic pain [106]. They discovered a clinically relevant difference between the treatment and placebo groups without statistical significance. They also discovered that it is a safe long-term management choice for MS with pain, since the number of side effects was comparable to that of normal treatment [106].

#### 2.2.3. Effects of Dronabinol on MS-Related Tremor

A total of 391 individuals in the CAMS study were given THC (Dronabinol), THC/CBD, or placebo orally; however, there was no statistically significant difference in the reaction to tremor as measured by a physician assessment (*p* = 0.052) or self-report (*p* = 0.398) [57].

#### 2.2.4. Effects of Dronabinol on MS-Related Bladder Functions

The CAMS trial [57] used oral cannabis extract or (Dronabinol) THC to evaluate bladder complaints in 667 participants with MS symptoms as a secondary outcome. There was no improvement in MS bladder symptoms when Dronabinol or oral cannabis extract were used [57].

#### 2.2.5. Effects of Dronabinol on MS-Related Sleep Disorders

The CAMS trial looked at the effects of marijuana on 630 MS patients with spasticity. For 15 weeks, MS patients were divided into four groups: oral Cannador, Dronabinol, Candor placebo, or Dronabinol placebo [57]. When compared to other placebo groups, the orally supplied group that took candor or Dronabinol demonstrated a significant improvement in MS sleep problems [57]. To assess the effect of Dronabinol on sleep difficulties, a case study was conducted on 52-year-old women with MS. Dronabinol was given orally at a dose of 2.5 mg BID for one week before being increased to 5 mg BID. The patient recorded the results using visual analog, concluding that the quality of sleep improved during the treatment [107].

#### 2.2.6. Effects of Dronabinol on MS-Related HRQoL

A three-week crossover study comparing the effect of Dronabinol on HRQoL in 24 MS patients found no significant differences in HRQoL in the treated patients compared to the placebo group [100].

#### 2.2.7. Effects of Dronabinol on MS-Related MS Disability and Disability Progression

Dronabinol’s effect on progression in progressive MS was studied in a randomized, double-blind, parallel, and placebo-controlled experiment [105]. For 36 months, patients were administered Dronabinol at a maximum dose of 28 mg daily or a placebo. In comparison to the placebo group, 35% of the Dronabinol group experienced at least one adverse event. In general, there were no safety issues or side effects associated with taking Dronabinol to reduce the course of MS [105].

### 2.3. Nabilone

Nabilone (Figure 1e) is the generic name for the primary synthetic analog of 9-THC carrying dibenzopyran-9-one structure with a racemic combination of (S,S)-(+)- and (R,R)-(−)-isomers. Nabilone was approved in 1985 for the treatment of chemotherapy-induced vomiting and nausea in individuals who did not respond to standard anti-emetic drugs; however, serotonin 5-HT3 receptor antagonists have partially supplanted Nabilone for this use. Nabilone is also approved for the treatment of neuropathic and chronic cancer pain, as well as MS spasticity. It comes in three dosages in pill form (0.25, 0.5, and 1 mg). The effectiveness of nabilone in the treatment of MS spasticity was investigated in three clinical studies published between 1995 and 2015 [108,109,110] and reviewed in four reviews published between 2006 and 2015 [40,44,46,111]. Nabilone appears to be beneficial in lowering most MS symptoms, including stiffness, pain, bladder dysfunction, and quality of life, according to the majority of trials (Table 1).

#### 2.3.1. Effects of Nabilone on MS-Related Spasticity

Martyan et al. studied the effects of Nabilone on MS-related spasticity for four weeks [108]. Their findings indicated that MS spasticity symptoms have improved [108]. A meta-analysis of three parallel groups based on the GRADE rating approach revealed that Nabilone has a moderate effect on MS-related spasticity [75].

#### 2.3.2. Nabilone Effects on MS-Related Pain

A meta-analysis of randomized clinical trials found sufficient evidence to support the improved impact of Nabilone on MS-related pain [75]. In 2015, a randomized double-blind study looked at the effects of Nabilone as an adjunctive to gabapentin for neuropathic pain caused by relapsing-remitting MS. It found that Nabilone combined with gabapentin is an effective, novel, and well-tolerated treatment for MS patients with neuropathic pain [110].

#### 2.3.3. Effects of Nabilone on MS-Related Tremor

Oral Nabilone was shown to have no meaningful effect on tremors in a class III trial [55]. In a systemic evaluation of 630 individuals, Koppel et al. concluded that oral Nabilone does not affect MS-related tremors [40].

#### 2.3.4. Effects of Nabilone on MS-Related Bladder Function

Martyn et al. observed significant relief of MS bladder dysfunction symptoms with oral Nabilone for two periods of four weeks in a double-blind crossover, placebo-controlled research [108].

#### 2.3.5. Effects of Nabilone on MS-Related Sleep Disorders

There was no clinical investigation that demonstrated the efficacy of Nabilone on the symptoms of MS sleep problems [72].

#### 2.3.6. Effects of Nabilone on MS-Related HRQol

In 2015, a randomized controlled trial looked at the efficacy of combining Nabilone and gabapentin for the treatment of MS-related neuropathic pain [110]. This combination was found to improve the quality of life for this group of people [110].

#### 2.3.7. Effects of Nabilone on MS Disability and Disability Progression

There was no clinical investigation that indicated how Nabilone affected MS-related disability and progression [72].

## 3. Adverse Effects

Several studies that employed cannabis in MS patients reported the side effects of the drugs [72,82,112,113]. The severity of these side effects was typically described as ‘mild’ to ‘moderate’, depending on a variety of factors, such as the dose, type, and quantities of cannabinoids in the products used, and the individuals [72,82,112,113]. Dizziness/lightheadedness (14–59%), gastrointestinal symptoms (diarrhea, nausea, and vomiting) (13–37% in active groups), dry mouth (4–26% in active groups), urinary tract infections (15.4–34%), and other adverse effects, such as fatigue, headache, attention disturbance, and disorientation, are the most commonly reported adverse effects in MS patients treated with cannabinoids [56,57,77,112]. A recent study tookadvantage of The Epistemonikos database, and analyzed 25 systematic reviews on the use of cannabinoids in MS patients; they concluded that the benefit–risk ratio of using cannabinoids in these patients is unfavorable, because there is a high level of evidence about the lack of benefits, and adverse effects are common [114]. The majority of assessments of these studies found no evidence that any of these side effects limit clinical use [72].

### 3.1. Nabiximols

In a class I research, 268 (46.9%) of 572 patients treated with Nabiximols reported at least one side effect, the most prevalent of which were dizziness and weariness [77]. The tolerability of Nabiximols was assessed in 325 patients in the MOVE 2 research, with dizziness, fatigue, drowsiness, nausea, and dry mouth reported in 16.6% of patients (54 out of 325 individuals) over three months [93]. The majority of these side effects (47 patients) were modest; however, there were significant side effects associated with Nabiximols, including despair, reduced walking ability, muscle spasms, and urinary tract infection [93]. In 11.4% of cases, the medicine was discontinued (37 subjects who have experienced adverse effects during the study period). The study classified it as well-tolerated [93]. Nabiximols were found to worsen balance control in MS patients in another study [115], particularly in multitasking conditions; however, a recent study found that Nabiximols had a short-term favorable effect on balance and walking abilities in MS patients [116]. In a recent cohort trial with 396 MS patients who were administered Nabiximols, 63 patients dropped out due to tolerance and safety concerns. The most commonly reported side effects were drowsiness (5.2%), vertigo/dizziness (4.8%), and weariness (3.8%). Despite this, no major side effects were noted during the titration and treatment stages of the trial [117].

### 3.2. Oral Cannabinoids and THC

The majority of the side effects were mild to moderate and dose-dependent, according to a review report that covered multiple clinical trials of CBD/THC in MS patients [46]. They stated that at least 10 mg of THC was required to treat spasticity; nevertheless, deleterious effects were identified at doses of 15 mg and higher [46].

While some studies have found that combining oral THC and CBD causes significantly more negative effects [63,97], others have found that the combination of THC and CBD causes fewer negative effects [57,118]. A total of 144 MS patients were treated with oral THC and CBD (titrated to a maximum dose of 25 mg THC daily) in a class I research, while 135 were given a placebo [112]. Adverse effects caused 30 out of 144 patients in the control group to quit out, while adverse effects caused 9 out of 135 patients in the placebo group to drop out [112]. Dizziness, dry mouth, weariness, urinary tract infection, asthenia, and headache were the most common reported adverse effects in the cannabis extract group, and they were all graded as mild to moderate [112]. In a meta-analysis assessment of cannabinoids’ tolerability in MS patients, the active treatment groups with dronabinol and cannabinoids, but not Nabilone, had a greater probability of withdrawal from the studies due to unpleasant effects [119]. Dizziness, dry mouth, and weariness were the most commonly reported side symptoms [119].

## 4. Conclusions

This study looks at the impact of various cannabis administration routes on MS patients with varying symptoms. It found indications that cannabis will support the efficacy of cannabinoids, namely throughan oromucosal spray and orally, in the treatment of pain and spasticity, which are the most common symptoms in MS patients. In general, adverse effects were modest to moderate, although special attention should be exercised in patients with multiple sclerosis.

## Figures and Tables

**Figure 1 life-12-00682-f001:**
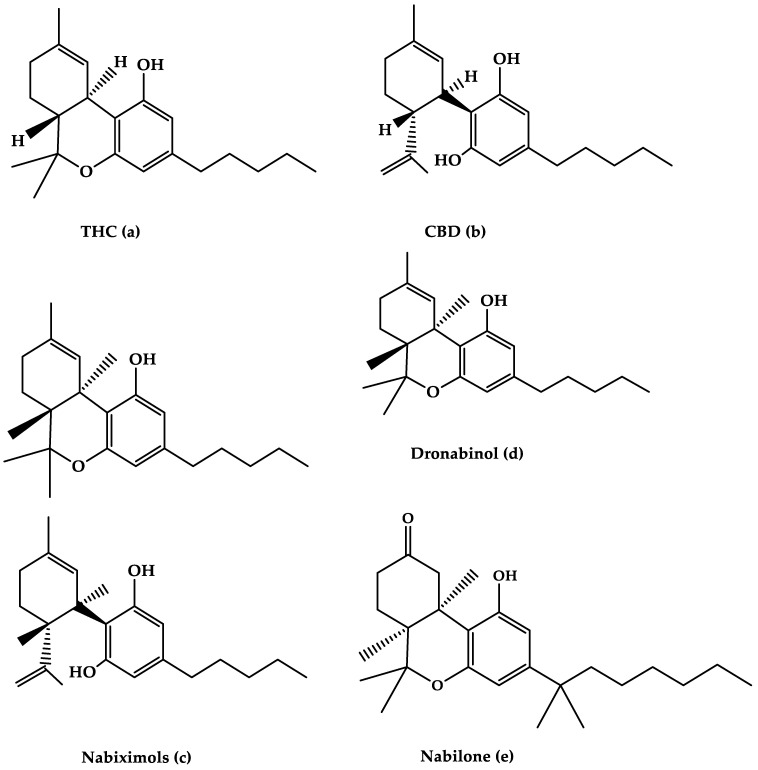
The molecular structures of (**a**) THC, (**b**) CBD, (**c**) Nabiximols, (**d**) Dronabinol, and (**e**) Nabilone are depicted in this diagram.

**Table 1 life-12-00682-t001:** The efficacy of oromucosal spray nabixiomol, oral dronabinol, and oral nabilone forms of cannabis on MS-related symptoms and their adverse effects.

Cannabinoid Agents	Therapeutic Actions	Adverse Effects
**Nabiximols**	Significant improvement in MS-related spasticity, pain, and sleep problems. Minor effect on the alleviation of bladder dysfunctions in MS patients and favorable effects but statistically insignificant effects onHRQoL. Unsuccessful in the treatment of MS-related tremors and ataxia and no significant changes in terms of disability and progression was shown in MS patients.	Moderate with the most commonly reported adverse effects: dizziness, fatigue, drowsiness, vertigo, and dry mouth.
**Dronabinol**	Highly effective in the treatment of pain in MS patients. Significant improvement in MS sleep problems. Other symptoms showed only little improvement or no effect.	Mild to moderate adverse effects: dizziness, dry mouth, and weariness.
**Nabilone**	Beneficial effects in lowering most MS symptoms, including spasticity, pain, bladder dysfunction, and quality of life. No significant effects on tremors. No clinical investigation on the symptoms of MS sleep problems and MS-related disability and progression.	Mild to moderateadverse effects: dizziness, dry mouth, weariness, and asthenia.

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
