# Peer review of "The Efficacy of Cannabis on Multiple Sclerosis-Related Symptoms"

_life, 2022, doi:10.3390/life12050682_

Round 1
Reviewer 1 Report
- Two chemical structures of Dronabinol were shown in Figure 1(d), is it redundant? or two associated forms ??, please confirm this part.
- Authors raise the topics " Cannabis To Treat Multiple Sclerosis Patients", that is vague, not very focusing.
- You raised five approved cannabis products in MS, I strongly recommend you should summarize your report of this review in form of tables including efficacy and adverse effects.
- The mechanisms the effects of cannabis in spasticity and pain is not emphasized, but you should discuss the parts, the distribution of the receptors of cannabis, CB1 CB2 etc.
Author Response
- Comment: Two chemical structures of Dronabinol were shown in Figure 1(d), is it redundant? or two
- Response: These two chemical structures are related to (C) Nabiximols as it contains a 1:1 molecular ratio of THC and CBD. Nabiximols is represented in the literature with these two chemical structures.
- Comment: Authors raise the topics " Cannabis To Treat Multiple Sclerosis Patients", which is vague, not very focusing.
Response: We change the title of the review to The Efficacy of Cannabis on Multiple Sclerosis -Related Symptoms
- Comment: You raised five approved cannabis products in MS, I strongly recommend you should summarize your report of this review in form of tables including efficacy and adverse effects
- Response: A table containing the efficacy of oromucosal spray nabixiomol, oral dronabinol, and oral nabilone forms of cannabis on MS-related symptoms and their adverse effects were added.
- Comment: The mechanisms the effects of cannabis in spasticity and pain is not emphasized, but you should discuss the parts, the distribution of the receptors of cannabis, CB1 CB2 etc.
Response: The introduction was modified to introduce the mechanisms of the effects of cannabis in spasticity and pain and the distribution of the receptors of cannabis, CB1 CB2

Reviewer 2 Report
The paper "Cannabis To Treat Multiple Sclerosis Patients" by Haddad and colleagues is a short review desribing evidence of Cannabis use in the treatment of Multiple Sclerosis and MS pain. I believe the review is well written, the topic is important, and the cited sources are comprehensive and updated. I like the structure of the review, which spans from a historical perspective to chemistry, pharmacology, medicine, molecular biology and epidemiology. Truly an engaging read, for which I thank the authors. The conclusions are well supported by the existing literature, and overall the manuscript is already excellent. I have only a few minor points, listed below. Note: all publications listed below are merely suggestions for expanding the paper and authors should not feel induced to include them specifically in their reference list.
- The authors do not mention the different properties of different Cannabis sativa strains, nor the differences between Cannabis indica and sativa, in the context of MS treatment. This is, I believe, a missing paragraph that would enrich the review. Many publications exist mentioning comparative analysis of Cannabis cultivars, strains and species, which could be cited in this review (e.g. https://www.frontiersin.org/articles/10.3389/fneur.2017.00299/full https://journals.lww.com/psychopharmacology/fulltext/2014/06000/therapeutic_satisfaction_and_subjective_effects_of.14.aspx )
- Another missing element is possibly the variability of MS patients, which is partly due to their genetic background (e.g. https://www.nature.com/articles/gene201134 https://www.ncbi.nlm.nih.gov/pmc/articles/PMC2848851/). The authors could briefly investigate if the usage of Cannabis would be more beneficial for certain subgroups of MS patients. If the answer is yes, this could open useful possibilities in the personalized treatment of MS.
- Antiplagiarism software detects a high similarity with this publication by Gado and colleagues: https://www.mdpi.com/2305-6320/5/3/91/htm and a low similarity with the publication by Uberall (https://www.ncbi.nlm.nih.gov/pmc/articles/PMC7027889/)
After analyzing both papers, I simply think the authors had in mind the previous ones when writing, and no intentional copy was done. However, I would avoid identical sentences such as "One dose equals one spray; the maximum dose per day is limited to 12 sprays, and the time between sprays should not..." and adjust the manuscript accordingly.
Author Response
We thank you for your comments which have been taken into consideration and addressed accordingly:
- Comment: The authors do not mention the different properties of different Cannabis sativa strains, nor the differences between Cannabis indica and Sativa, in the context of MS treatment. This is, I believe, a missing paragraph that would enrich the review. Many publications exist mentioning comparative analysis of Cannabis cultivars, strains, and species, which could be cited in this review (e.g. https://www.frontiersin.org/articles/10.3389/fneur.2017.00299/full https://journals.lww.com/psychopharmacology/fulltext/2014/06000/therapeutic_satisfaction_and_subjective_effects_of.14.aspx ).
- Response: A new paragraph has been added, which describes the comparative analysis of Cannabis species, and the recommended publications were cited.
- Comment: Another missing element is possibly the variability of MS patients, which is partly due to their genetic background (e.g. https://www.nature.com/articles/gene201134 https://www.ncbi.nlm.nih.gov/pmc/articles/PMC2848851/). The authors could briefly investigate if the usage of Cannabis would be more beneficial for certain subgroups of MS patients. If the answer is yes, this could open useful possibilities in the personalized treatment of MS.
- Response: A new paragraph has been added, which describes the variability of MS patients. Regarding if the usage of Cannabis would be more beneficial for certain subgroups of MS patients, the answer is still unknown as Furgiuele et al reported in their review in 2021, more studies are needed to further understand the pharmacogenetics of MS disease and to define the molecular target of CBD in MS patient by performing the ex vivo/in vitro research in human immune cells. This has been added as a paragraph to our review.
- Comment: Antiplagiarism software detects a high similarity with this publication by Gado and colleagues: https://www.mdpi.com/2305-6320/5/3/91/htm and a low similarity with the publication by Uberall (https://www.ncbi.nlm.nih.gov/pmc/articles/PMC7027889/)
After analyzing both papers, I simply think the authors had in mind the previous ones when writing, and no intentional copy was done. However, I would avoid identical sentences such as "One dose equals one spray; the maximum dose per day is limited to 12 sprays, and the time between sprays should not..." and adjust the manuscript accordingly.
- Response: The similar sentence has been paraphrased.

Round 2
Reviewer 1 Report
The authors create an essential table in this version, the content had been retified with good satification. I suggest to accept this revised manuscript.